# Physiological Investigation and Transcriptome Analysis Reveals the Mechanisms of *Setaria italica*’s Yield Formation under Heat Stress

**DOI:** 10.3390/ijms25063171

**Published:** 2024-03-09

**Authors:** Manicao Hu, Meng Yang, Jingyang Liu, Haozhe Huang, Ruiwei Luan, Hongliang Yue, Caixia Zhang

**Affiliations:** College of Agronomy, Shandong Agricultural University, Taian 271018, China; manicaohu0908@163.com (M.H.); ym198206@163.com (M.Y.); ljy076189@163.com (J.L.); haozhehuang2001@163.com (H.H.); ruiweiluan0428@163.com (R.L.); hl_yue2002@163.com (H.Y.)

**Keywords:** *Setaria italica*, heat stress, sugar transport, yield formation, SUT, SWEET

## Abstract

*Setaria italica* is an important crop in China that plays a vital role in the Chinese dietary structure. In the last several decades, high temperature has become the most severe climate issue in the world, which causes great harm to the yield and quality formation of millet. In this study, two main cultivated varieties (ZG2 and AI88) were used to explore the photosynthesis and yield index of the whole plant under heat stress. Results implied that photosynthesis was not inhibited during the heat stress, and that the imbalance in sugar transport between different tissues may be the main factor that affects yield formation. In addition, the expression levels of seven *SiSUT* and twenty-four *SiSWEET* members were explored. Sugar transporters were heavily affected during the heat stress. The expression of *SiSWEET13a* was inhibited by heat stress in the stems, which may play a vital role in sugar transport between different tissues. These results provide new insights into the yield formation of crops under heat stress, which will provide guidance to crop breeding and cultivation.

## 1. Introduction

The global temperature has increased continuously in the last few decades, and high temperature weather occurs frequently [1,2,3]. Many places experienced their highest temperature in 2023, a year that made history. In addition to causing droughts and fueling fires, extremely high temperature has seriously affected food security. Heat stress is a major abiotic factor that dramatically limits plant growth and yield [4,5,6,7,8]. Hence, heat damage is a major issue for the crop industry and the whole community, particularly given the recent and continuous threat of global warming.

Among the key climate change drivers, high temperatures influence all growth stages during rice’s life cycle [9,10,11]. Research has shown that short-term or long-term stress resulting from high temperatures can cause a series of morphological, physiological, biochemical, and molecular changes in crops and result in irreversible changes that limit cell division and growth, reduce fertility, promote senescence, and even, in extreme cases, cause cell death [12,13,14,15].

*Setaria italica* (millet) is the main crop of dry ecological agriculture, a strategic reserve crop to cope with climate change, and the model crop of functional gene research in the new era. Its seeds are rich in protein, fat, and vitamins and have high nutritional value and relatively balanced nutrition. Millet plays an important role in improving the food nutrition structure and food security [16]. At the same time, the nutritive organs of millet contain rich nutrients and are crucial in animal feeding. The yield and quality of *S. italica*, as a traditional crop with strong stress resistance, are also affected by extreme high temperature weather [17,18]. Heat stress is a major factor that limits crop biomass, seed-setting rate, grain yield, and quality, and also exerts a transgenerational effect, whose mechanistic basis is associated with epigenetic regulation [19]. Research has also shown that the anthesis stage in crops is highly sensitive to high temperatures [20]. During this period, high temperature stress leads to the obstruction of photoassimilate transport, a decreased grain filling rate, and poor filling [14,15,21]. A spikelet tissue temperature of 33.7 °C for an hour that coincides with anthesis has been documented to be sufficient to induce spikelet sterility [22], and it has been found that exposure to temperatures of 38 °C or 41 °C for an hour after anthesis does not induce sterility [23].

In recent years, extremely high temperatures often occurred at the grain filling stage, leading to poor grain filling and reduced grain weight, which seriously affected crop yield [24,25,26,27]. Grain filling is a process where sugars are transported from the source organ (including photoassimilates produced by leaves and nonstructural carbohydrates stored in the stems and sheaths) to the grain through the phloem (in the form of sucrose) to complete grain filling [15]. Hence, sugars synthesized in the photosynthetic cells are ultimately exported to the apoplasm (cell wall), most likely by Sugar Will Eventually be Exported Transporters (SWEET) efflux proteins, before entering into the sieve elements–companion cells (SE–CC) complexes via sucrose transporters (SUT) [28,29,30]. Previous studies have shown that sucrose transporters play a regulatory role in the head and end regulation of phloem by regulating the physiological activities of source and sink [31]. Additionally, this process is crucial for plant tolerance of abiotic stress.

Crops are more sensitive to stress at the reproductive growth stage, especially the grain filling stage, than to stress at the vegetative growth stage [32,33]. The high sensitivity to stress in this period is related to the downstream stress response caused by the disturbance of sucrose metabolism and the reduction in hexose content [30]. In a high temperature environment, panicle temperature if higher than leave temperature, and high temperature inflicts greater damage on the panicles than on the stems, sheaths, and leaves. On the condition that photosynthesis is not impaired, grain growth is considerably inhibited, which directly affects yield formation [34]. Relevant studies have proven that the ability to transport carbohydrates to developing grains determines the size and number of grain cells [35]. At the grain filling stage, and especially the early stage, water and nutrients accumulate rapidly in the grains, resulting in volume and weight increments [14]. If high temperature events occur during this period, the transport of sugars from the source to the sink organ will be greatly reduced, and grouting will decrease or even stop because of insufficient sugar supply [36]. Grain number and weight will also decrease remarkably, ultimately causing irreversible yield loss [18,37].

Considerable proportions of blighted grains are produced every year, but the photosynthetic rate of leaves does not decrease under high temperature, indicating that storage capacity and source activity are not factors that limit yield formation. The dry matter weights of the stem sheaths and leaves increase, and the dry matter weight of the panicles decrease under high temperature conditions. Therefore, the authors speculate that an insufficient supply of photoassimilates transported from the source organs to the sink grains may be the key factor in the decrease in grain weight. However, current research of *S. italica* is incomplete. In particular, research of the mechanisms of the effects of high temperature stress on the formation of *S. italica* yield is found to be insufficient when compared to studies of rice and other major grain crops.

Thus, in this study we used two main cultivated varieties, namely Zhonggu2 (ZG2) and Ai88 (AI88), to investigate photosynthesis, dry matter weight, and carbohydrate accumulation and distribution in the whole plant under heat stress. The members of sugar transport families in the genome were identified, and the key genes that could regulate yield formation were also determined through RNA sequencing (RNA-seq) and an analysis of the expression level of sugar transporters, which clarified how heat stress affects the source–sink relationship in *S. italica* at the early stage of grain filling [38]. This study provides new insights into yield formation during heat stress and also provides new ideas about heat resistance breeding and cultivation.

## 2. Results

### 2.1. Yield, Photosynthetic Rate, and Chlorophyll Content of the Two Varieties under Heat Stress

The indices of photosynthetic rate, grain number per panicle, seed setting rate, and grain weight were investigated to explore the main factor of yield formation during heat stress. The chlorophyll content and net photosynthesis speed were used to evaluate photosynthetic capacity under heat stress. The results obtained for chlorophyll content indicated that heat stress had no remarkable effect on the ZG2 and AI88 plants. In AI88, chlorophyll content increased after heat stress, but not considerably (Figure 1a). Heat stress had little effect on the net photosynthetic rate of the ZG2 and AI88 plants (Figure 1b). But the yield of ZG2 and AI88 plants were affected remarkably by heat stress, as decrements of 14.08% and 22.88% were recorded for the ZG2 and AI88 plants, respectively, when exposed to heat stress and compared with the corresponding control (Figure 1c). In order to assess the level of stress in plants used in experiments, we investigated the Chl a/Chl b ratio of flag leaves, and recorded the malonaldehyde (MDA) content of leaves, the sheaths and stems, and panicles. The Chl a/Chl b ratio of flag leaves increased under heat stress, especially in AI88 plants, as the ratio of ZG2 were 2.69 (control) and 2.97 (heat stress), and the ratio of AI88 were 3.10 (control) and 5.14 (heat stress) (Appendix A). The MDA content increased in the flag leaves, sheaths and stems, and panicles, and the difference between control and heat stress were, except for the leaves of ZG2 plants (Appendix A), found to be significant.

Yield formation factors are shown in Appendix A, in which the grain number is seen to be reduced by heat stress in both genotypes, when compared with the corresponding control, with larger decrements being found in ZG2 than in AI88. Decrements of 25.43% and 8.73% were recorded for the ZG2 and AI88 plants, respectively (Appendix A). The effect of heat stress on the seed setting rate of the ZG2 and AI88 plants was less than the effect on the kernel weight and grain number, and high temperature had no effect on the seed setting rate of both ZG2 and AI88 plants (Appendix A). And the filled grain number decreased significantly in both ZG2 and AI88 plants (Appendix A). The kernel weight of ZG2 was only affected slightly by heat stress, but was considerably decreased by heat stress in AI88, as indicated by the 9.71% reduction in the AI88 plants (Appendix A).

### 2.2. Accumulation and Allocation of Dry Matter Weight under Heat Stress

The percentage of dry matter weight in the panicles of both varieties decreased at harvest, compared with the dry matter weight of the whole plant. Decrements of 9.53% and 14.09% were recorded for the ZG2 and AI88 plants under heat stress, respectively, compared with the corresponding controls (Figure 2a). No significant changes in the percentage of the dry matter weight of the leaves were observed, compared with the total dry matter weight in ZG2 and AI88 (Figure 2b). In comparison, a remarkable increase in the percentage of the dry matter weight of the sheath stems, relative to that of the total dry matter weight, was found in AI88 under heat stress, when compared with the controls. And a significant increase was observed between the control and heat stress in ZG2, as increments of 9.82% and 20.17% were recorded for the ZG2 and AI88 plants, respectively, under heat stress, when compared with the corresponding controls (Figure 2c).

### 2.3. Content of NSC, Soluble Sugar, and Starch in Grains, Leaves, and Sheath Stems under Heat Stress

The content of non-structure carbohydrate (NSC) in the grains was reduced by heat stress in both genotypes, compared with the corresponding control, and larger decrements were found in AI88 than in ZG2. Decrements of 7.57% and 16.42% were recorded for the ZG2 and AI88 plants, respectively, under heat stress, compared with the corresponding control (Figure 3a). In contrast, increases in the NSC content were observed in the leaves under heat stress, and larger increases were found in AI88 than in ZG2 (Figure 3b). Heat stress had a notably different effect on the NSC content of the ZG2 and AI88 plants when compared with the control in the sheath stems. In the AI88 plants, the NSC content of the sheath stems were considerably increased by heat stress, as indicated by the 22.42% and 61.29% increase found in the ZG2 and AI88 plants, respectively (Figure 3c).

The total soluble sugar content of the panicles was not affected by heat stress in both genotypes (Figure 3d). The changes in the content of total soluble sugars differed in the leaves of the two genotypes under heat stress. A significant decrease in the total soluble sugar content was found in the ZG2 leaves, but a clear increase was found in the AI88 leaves (Figure 3e). Meanwhile, a notable difference in the total soluble sugar content was observed in the sheaths and stems of ZG2, and a higher increase was found in the sheaths and stems of AI88 under heat stress, as indicated by the 62.14% increase observed in the AI88 plants, compared with the control (Figure 3f).

With regard to starch, its content in the grains of ZG2 plants was considerably decreased by heat stress, compared with control conditions, and a notable decrease was found in AI88 plants under heat stress (Figure 3g). A notable difference in starch content was found in the leaves and sheath stems of the control and heat-stressed plants of ZG2 and AI88. Sharp increases in leaves and sheath stems were found in ZG2 plants, when compared with the corresponding controls. And remarkable increments were observed in leaves and sheath stems of the AI88 plants under heat stress, as evidenced by the increments of 43.08% and 60.40% recorded in the leaves and sheath stems, respectively, of the AI88 plants under heat stress, when compared with the corresponding controls (Figure 3h,i).

### 2.4. RNA-Seq Analysis of Heat Stress in Three Tissues

In this study, RNA-seq was used to explore the differentially expressed genes (DEGs) during heat stress in three tissues of the two varieties. In total, 1.05 G reads were obtained, with an average of 8.75 Gb for each sample. The RNA-seq datasets were mapped onto the *S. italica* genome (V2.2), and the average map rate was approximately 96% (Appendix A). On average, 18,406 (ZG2) and 18,125 (AI88) genes were expressed in the two varieties; the expressed genes in panicles were highest (ZG2:20,516, AI88:19,745), and leaves had the least expressed genes (ZG2:16,422, AI88:16,268). The study then proceeded to obtain 2743 (ZG2, leaves), 3085 (AI88, leaves), 2533 (ZG2, stems), 4092 (AI88, stems), 1706 (ZG2, panicles), and 2111 (AI88, panicles) DEGs (Figure 4). The number of DEGs in AI88 was higher than that in ZG2 in all the tissues, and there were fewer DEGs in the panicles than in the two other tissues. Venn analysis showed that 1647 (leaves), 1591 (stems), and 571 (panicles) DEGs were shared by the two varieties (Figure 5a–c). The expression pattern of these DEGs were highly consistent in the different varieties (Figure 5d). In addition, the function annotation of these genes revealed many genes related to long-distance transport of sugars in plants, including SWEETs and SUTs. GO enrichment also indicated that transmembrane and other transporters were significantly enriched in all samples. The transmembrane transport genes were therefore heavily affected during the heat stress.

### 2.5. Identification of Sugar Transport-Related Genes in S. italica Genome

After it was observed that the dry matter distribution and functional annotation of the DEGs suggested that sugar transport may be affected by high temperature stress, two major sugar transport-related families in *S. italica* were identified. In total, seven *SiSUT* and twenty-four *SiSWEET* members were identified in the genome (Appendix A), which we named in accordance with the orthologous genes in *Oryza sativa* (Figure 6a,b and Appendix A, and Appendix A). Phylogenetic and gene structure analysis indicate that gene structure similarity in the SUT family was very low; however, in the SWEET family, members with closer evolutionary relationships were found to have similar gene structures (Figure 6), which implied SWEETs could possibly have experienced recent duplication events.

Gene loci analysis of both families indicated that these genes were located in eight of the nine chromosomes, and only chromosome 7 did not contain these family members (Figure 7a). The segment duplication analysis indicated that all these members experienced complex duplication events. The SUT family members were involved in 11 segment duplication events, and no tandem duplication events were identified in this family (Figure 7b). In the SWEET family, we identified two tandem repeat events (*SiSWEET4a-c* and *SiSWEET3d-e*). Aside from the tandem repeats, 67 segment duplication events related to the expansion of these families were also identified (Figure 7b). All these duplication events contribute to the expansion and formation of these two families.

### 2.6. Expression Pattern of SUTs and SWEETs under Heat Stress

Among the sugar transport genes, two *SUTs* and twelve *SWEETs* were expressed differently in three tissues during the heat stress treatment (Appendix A). Detailed analysis of these DEGs showed that the expression patterns varied in the different tissues. In the leaves, nine genes were differentially expressed after heat treatment. *SiSWEET6, SiSWEET4c, SiSWEET16a*, *SiSUT1*, and *SiSUT5a* were differentially expressed during heat stress in both varieties. *SiSWEET1a* and *SiSWEET16b* were only expressed differentially in ZG2, and *SiSWEET4a* and *SiSWEET15* were only expressed differentially in AI88 (Figure 8). In the stems, seven genes were identified as being considerably differentially expressed. Only two genes, *SiSWEET15* and *SiSWEET13a*, were differentially expressed in both varieties, and all seven DEGs were found in the AI88 stem (Appendix A).

The expression level of these sugar transport DEGs indicated that five sugar transport-related DEGs in ZG2 decreased during heat stress in the panicles, but only *SiSWEET14a* was identified as a DEG in AI88 panicles, with this gene increasing during heat stress in AI88 (Figure 8) and decreasing in ZG2. In the leaves, seven sugar transporters were differentially expressed in ZG2 during heat stress, of which two (*SiSWEET6* and *SiSWEET16b*) were more highly expressed than the controls, and five (*SiSWEET4c*, *SiSWEET1a*, *SiSWEET16a*, *SiSUT1*, and *SiSUT5a*) showed decreased expression during heat stress (Figure 8). In AI88 leaves, only the expression level of *SiSWEET6* increased during heat stress, and six other genes (namely *SiSWEET4a*, *SiSWEET4c*, *SiSWEET16a*, *SiSWEET15*, *SiSUT1*, and *SiSUT5a*) decreased during heat stress. Hence, the expression level of *SiSWEET6* increased and *SiSWEET4c*, *SiSWEET16a*, *SiSUT1*, and *SiSUT5a* decreased in both varieties of leaves during heat stress (Figure 8).

In the sheath stems, two sugar transporters, namely an upregulated *SiSWEET15* and a down regulated *SiSWEET13a*, were identified as DEGs in ZG2. In AI88, seven genes were identified as DEGs. Only *SiSWEET13a* and *SiSWEET13b* considerably decreased during heat stress. Sugar transport genes were briefly heavily affected during the heat stress, which will affect the sugar transport efficiency and the whole plant’s dry matter partition.

## 3. Discussion

### 3.1. Photosynthesis Is Not the Direct Means of Yield Decrease during Heat Stress

The chlorophyll content and net photosynthesis rate of this study indicated that heat stress had little adverse effect on photosynthesis; the chlorophyll content even increased during heat stress in AI88, indicating chlorophyll content was not the main reason for the yield decrease under the high temperature. This result was consistent with those obtained by previous studies of other plants [14,39]. The main physiological indices of photosynthesis were not affected by heat stress, and some indices even increased during the treatment, which could not cause irreversible damage to the photosynthetic machinery of leaves [31]. Light capture capability or net photosynthetic rate are not therefore the direct factor that leads to yield decrease during heat stress [12,14,40].

Regarding the yield of ZG2 and AI88, plants were remarkably affected by heat stress, and decrements of 14.08% and 22.88% were recorded, respectively (Figure 1c). Our results showed that the kernel weight of ZG2 was only affected slightly by heat stress but was considerably decreased by heat stress in AI88, with a sharp decline in grain number in both genotypes, compared with the corresponding control. Results of 25.43% and 8.73% were recorded for the ZG2 and AI88 plants, respectively. The effect of heat stress on the seed setting rate of the ZG2 and AI88 plants was less than the effect on the kernel weight and grain number, and high temperature had no effect on the seed setting rate of both ZG2 and AI88 plants (Appendix A), which suggested that the decrease in grain number and kernel weight might be the main factors responsible for the lower yield in AI88 under heat stress, and that the decrease in grain number might be the main factor responsible for the lower yield in ZG2. The stable or increased seed setting rate, meanwhile, could be caused by the falling of the abnormal grains caused by heat stress [41]. Crop seed development and yield formation are the sugars transported from photosynthetic leaves into grains, and the imbalance of sugar transport always caused yield decline [42,43,44]. We therefore investigated the photoassimilate transport under heat stress.

### 3.2. Sugar Transport Is One of the Main Factors That Influence Yield

According to the dry matter distribution, high temperature led to a weight increase in the leaves and stems, but the panicles’ weight decreased remarkably (Figure 2). Evidently, heat stress influenced the sugar transport during heat stress [45]. The indices of the seeds and panicles also indicated that, during heat stress, the lack of sugar transported from the leaves through apoplastic and symplastic pathways heavily affected the development of the seeds and led to a decrease in yield and quality. All these indices imply that the imbalance in sugar transport led to the weight increase in the leaves and stems but limited the development of the panicles. Accordingly, a higher decline in panicles’ dry matter weight and a higher increase in sheath stems were found in AI88 than in ZG2 plants, when both were under heat stress (Figure 2a,c). A similar pattern of changes was found in the NSC and starch content in panicles and sheath stems of plants under heat stress (Figure 3). These results indicated that fewer photoassimilates were available in panicles, and that insufficient sucrose unloading into the grains caused the kernel weight to decrease under heat stress, especially in those with heat sensitive cultivars [46]. This finding indicated that the disorder of the distribution of dry matter caused the decline in yield. And heat stress at the grain-filling stage could directly impact carbon utilization and distribution in plants, especially those with heat sensitive cultivars [46,47].

Sucrose is the primary organic carbon transported through the phloem from photosynthetic leaves (source) into non-photosynthetic tissues (sink) [48]. In addition, RNA-seq of the three tissues confirmed that the expression levels of the sugar-related genes and sugar transporters were heavily influenced during heat stress. Further analysis confirmed that 50% of the SWEETs genes that participated in long-distance sugar transport were differentially expressed after high temperature treatment. Hence, the influence of sugar transport may be one reason for the imbalanced distribution of dry matter [14,49].

### 3.3. Sugar Transporters Played Important Roles in the Sugar Transport under Heat Stress

As mentioned above, a higher increase in the dry matter weight was found in sheath stems than in leaves under heat stress, when compared with their respective controls (Figure 2). This finding indicated that heat stress directly impaired carbon utilization and distribution in plants, especially in sheath stems. The expression pattern of SUTs and SWEETs families also confirmed that sugar transport was heavily affected during the heat stress. The stem is the main tissue that transports sugars from the leaves to the panicles, so the sugar transporters in the stems are crucial in plant yield formation. In this study, only two transporters, *SiSWEET13a* and *SiSWEET15*, were differentially expressed in the stem during heat stress (Figure 8). The expression level of the two genes implied that only the expression level of *SiSWEET13a* was inhibited during heat stress, and that this inhibition was correlated with the decline in transportation capacity in the stems during heat stress. Studies of Arabidopsis have confirmed that At *SWEET13a*/*b*/*c* mainly works by pushing sugar to the extracellular transport pathway [50]. Thus, during heat stress, the sugar pushed to the extracellular transport pathway is limited because to the suppression of *SiSWEET13a*, and less sugar is transported, whereas a part of sugars was left in the sheath stems or leaves, leading to an increase in the weight of the leaves and stems and a decrease in grains. The detailed mechanism of how *SiSWEET13a* influences sugar transport still needs further research. In addition, *SiSWEET13a*, *SiSWEET13b* were also detected, and only seen to decrease in AI88. Taking into account the similarity of these two genes, this decrease could possibly exacerbate yield decrease during heat stress in AI88.

In addition, many varieties of specific DEGs were also identified in this study, including *SiSWEET14a* and *SiSWEET14b*. In this study, the expression level of *SiSWEET14a* in AI88 panicles increased during the heat stress, while the expression of *SiSWEET14b* decreased in the ZG2 panicle. According to the study of *OsSWEET14*, this gene was mainly related to disease resistance [51,52,53,54,55]. The reduction of *OsSWEET14* could lead to plant height increase, but without producing reduced yield [52,55]. Over-expression of *OsSWEET14* could lead to the decline of kernel weight and grain number [52]. All these results indicated that *OsSWEET14* was a negative regulatory factor of the yield. These results were in accordance with the physiological data of our studies, and also partly explain the difference of the yield formation in ZG2 and AI88.

## 4. Materials and Methods

### 4.1. Experimental Setup and Plant Materials

This study was conducted in the experimental farm of Shandong Agricultural University, Taian, Shandong Province, China. Two main *S. italica* cultivars, namely ZG2 (heat-tolerant) and AI88 (heat-sensitive), with different heat tolerance were used. The foxtail millet seeds were sown in pots (diameter 35 cm, height 32 cm) filled with soil. In this study, there were four groups, including control of ZG2, heat stress of ZG2, control of AI88, and heat stress of AI88. Each group had fifteen pots, which each contained five plants. The plants were cultivated and irrigated, as in normal cultivation, before the early stage of grain filling. The plants were then divided into control and heat stress groups and transferred to two greenhouses with automatic temperature control system in the grain filling stage. Thereafter, *S. italica* plants were acclimated for about two days in the artificial climate chamber (60–70% relative humidity, 29–31 °C/23–25 °C day/night temperatures) and divided into two groups. On the following day, one group was subjected to a heat stress treatment (60–70% relative humidity, 39–41 °C from 8:30 am to 4:30 pm, 28–30 °C nights for 15 days), and the other group served as the control (60–70% relative humidity, 29–31 °C days and 23–25 °C nights). After 15 days of heat stress, all plants were removed from the artificial climate chambers and cultivated, as in normal cultivation, until harvest.

On Day 15 of the heat stress, the flag leaves, sheath stems, and panicles were sampled to determine the concentration of carbohydrates. At the same time, the materials of the flag leaves, stems and panicles were collected from the plant, frozen with liquid nitrogen, and stored at −80 °C. The materials were then processed through RNA-seq, in accordance with the standard protocol of Berry Genomics Company (Beijing, China). The sequence was processed with the Illumina NovaSeq5000 platform (model PE150). In this study, three plants were selected randomly as one measure repeat, and three measurement repetitions were processed in each group. At the same time, three repetitions for each determination were carried out in the experiment.

### 4.2. Measurement of Net Photosynthetic Rate, Chlorophyll Concentration, MDA Content, and SeedSetting Rate

On Day 15 of the heat stress treatment, the chlorophyll concentration in the flag leaves was determined, and the net photosynthetic rate was analyzed. The net photosynthetic rate of the flag leaves was measured with a Li-Cor 6400 portable photosynthesis system (Li-Cor Inc., Lincoln, NE, USA). Six randomly selected plants were used for the data acquisition of each group, and three technical repetitions were used for each determination. The photosynthetic photon flux density was set to 1200 μmol m^−2^ s^−1^, the leaves’ area was set to 6 cm^2^, the flow speed was set to 500 μmol s^−1^, and temperatures were set to 30 °C and 40 °C.

For the measurement of chlorophyll concentration, MDA content, and the seedsetting rate, three randomly selected plants were used as one repeat, and five measurement repetitions were processed in each group. Three measurement repetitions for each determination were also carried out. Chlorophyll was extracted according to the method described by Sartory and Grobbelaar [56] and Li [13]. About 0.1 g flag leaves cut into pieces were immersed in 20 mL 95% (*v*/*v*) ethanol, and chlorophylls was extracted in ethanol for 48 h under dark conditions. The concentration of total chlorophyll was determined by measuring the absorbance, at 665 (D665), and 649 nm (D649), with a spectrophotometer (Lambda 25, Perkin Elmer, Freemont, CA, USA). The content of chlorophyll a (*C*_a_) and chlorophyll b (*C*_b_) was calculated by the formulas: *C*_a_ (μg·mL^−1^) = 13.95 × D665 − 6.88 × D649, *C*_b_ (μg·mL^−1^) = 24.96 × D649 − 7.32 × D665, and the Chl a/Chl b ratio were then analysed.

The MDA content was estimated by measuring the concentration of thiobarbituric acid reactive substances (TBARS). Frozen flag leaves, sheaths, stems and panicles (0.5 g) were homogenized in 2 mL of 5% trichloroacetic acid, before being centrifuged at 10,000× *g* for 15 min. A total 2 mL supernatant, 1 mL 20% (*v*/*v*) TCA, and 0.5% (*w*/*v*) thiobarbituric acid was then added to make the reaction mixture. The reaction mixture was incubated in a water bath at 100 °C for 30 min, and the absorbance was then measured at 450, 532 and 600 nm, respectively. The MDA content was calculated using the following formula: C (μM) = 6.45(A532 − A600) − 0.56A450.

At maturity, millet plants were harvested and dried to a constant weight. They were then threshed by hand, before the seed setting rate was determined by counting the number of filled seeds and all of the seeds in each panicle. The thousand kernel weight was calculated by measuring the number of filled seeds and obtaining their weight.

### 4.3. Analysis of Accumulation and Allocation of Dry Matter Weight

At harvest, the *S. italica* plants were sampled to determine their dry matter weight and distribution. The plants were divided into panicles, leafs, and sheath stems, before being inactivated at 105 °C for 30 min, and then dried at 85 °C for 48 min.

### 4.4. Assays of Nonstructural Carbohydrate, Soluble Sugar, and Starch Contents

The flag leaves, sheath stems, and panicles were sampled, to determine the concentrations of total soluble sugar, nonstructural carbohydrate (NSC) content, and starch content after 15 days of heat stress.

A total of 0.2 g of spikelets, flag leaves, and sheathstems (dry matter weight) were homogenized with 10 mL of deionized water and boiled for 30 min three times. The extract was gathered and diluted with deionized water to a volume of 50 mL, and then mixed. The extracting solution was used to measure the total soluble sugar content. The content was determined with the method of [38], with some modifications. The total soluble sugar content was determined using the sulfuric acid anthrone colorimetric method, and the absorbance was measured at 620 nm. The starch was extracted from the sediment, which was dried, weighed, and boiled in deionized water with 9.2 and 4.6 M perchloric acid. The supernatant was then used to determine the starch content by applying the sulfuric acid anthrone colorimetric method. The total NSC was considered the sum of the content of soluble sugars and starch.

### 4.5. RNA-Seq Analysis

The RNA sequenced datasets were first cleaned with Trimmomatic [57], with a minimum reads length of 90 bp, before the cleaned reads were mapped onto the reference genome of *S. italica* (v2.2, https://phytozome-next.jgi.doe.gov (accessed on 10 August 2022)) by using Tophat2 [58] with the default parameters. The expression level was obtained with Cufflinks [59]. The differentially expressed genes (DEGs) were identified with Cuffdiff [59] (only the reference transcript annotation were considered in this study), and the significant DEGs were selected in accordance with the parameter of FDR < 0.05 and |log2 (change fold)| > 1.

### 4.6. Identification of SWEET and SUT Members

The hidden Markov model (HMM) model of SUT was constructed with the well-aligned SUT sequences from *Oryza sativa* and *Arabidopsis thaliana* by HMMBUILD [50]. The model of SWEET was downloaded from Pfam (http://pfam.xfam.org (accessed on 9 October 2022)) with the domain of MtN3_slv (PF03083). The members were first identified by HMMSEARCH [60] with the HMM model and the protein sequences of *S. italica*. BLASTP [61] was also utilized to identify the homolog genes of these families. All the candidate genes were inputted into CDD search (https://www.ncbi.nlm.nih.gov/Structure/cdd/wrpsb.cgi (accessed on 12 October 2022)) [62], and only the genes that contained the domain were retained as the family members.

The neighbor joining trees of the SWEET and SUT family were constructed based on the protein sequence of the family members, and MEGA11 [63] was used to process this construct with a bootstrap of 1000. Segment duplication was processed through a whole genome duplication analysis of the millet genome by MCScanx [64] with the parameter of “-s 5 -e 1e-5 -m 25 -w 5”. Duplication events containing SUT or SWEET family members were selected and analyzed [65].

### 4.7. Statistical Analyses

Data were processed with SPSS 11.5 for Windows. The mean value and standard errors in the figures represent three replications, unless otherwise stated. The differences between treatments and genotypes were compared by using Tukey’s least significant difference (LSD) at the 5% probability level.

## 5. Conclusions

The significant decline in yield caused by heat stress is due to the assimilate distribution rather than photosynthesis, as no significant difference in the photosynthesis of the leaves were noted. Heat stress caused a notable decrease in the dry matter weight of panicles, while the comparable measure for sheath stems clearly increased, particularly in the AI88 plants. Heat stress could limit the expression level of long-distance sugar transporters in millet, and this inhibition could lead to sugar transport imbalance and yield reduction. These results provide new insights into yield formation during heat stress that will contribute new guidance to heat stress breeding and cultivation.

## Figures and Tables

**Figure 1 ijms-25-03171-f001:**
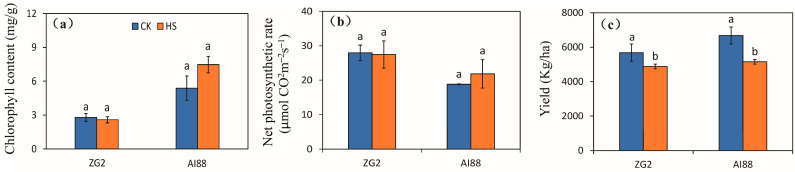
Effect of heat stress on the chlorophyll content, net photosynthesis rate, and yield. (**a**) The chlorophyll content in flag leaves of ZG2 and AI88, respectively; (**b**) Net photosynthesis rate in flag leaves of ZG2 and AI88; (**c**)Yield of ZG2 and AI88. A *t*-test was conducted for data to compare the difference between control and heat stress within one cultivar. CK, control; HS, heat stress. Different bars and low cased letters within the column chart indicate statistically significant differences between different treatments applied to the same cultivar (*p* < 0.05).

**Figure 2 ijms-25-03171-f002:**
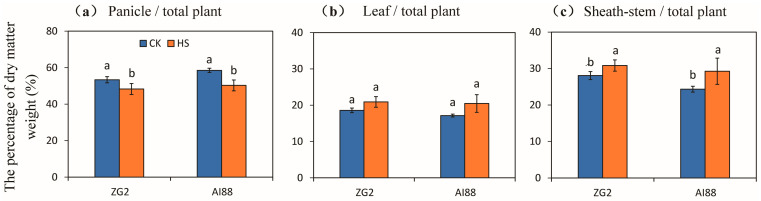
Effect of heat stress on the dry matter weight accumulation and distribution in the panicles, leaves, and sheathstems of millet. The percentage of dry matter weight of the (**a**) panicles, (**b**) leaves, and (**c**) sheathstems were compared to the whole plant of ZG2 and AI88, respectively. CK, control; HS, heat stress. A *t*-test was conducted on data to compare the difference between control and heat stress within one cultivar. Different bars and low cased letters within the column chart indicate statistically significant differences between different treatments applied to the same cultivar (*p* < 0.05).

**Figure 3 ijms-25-03171-f003:**
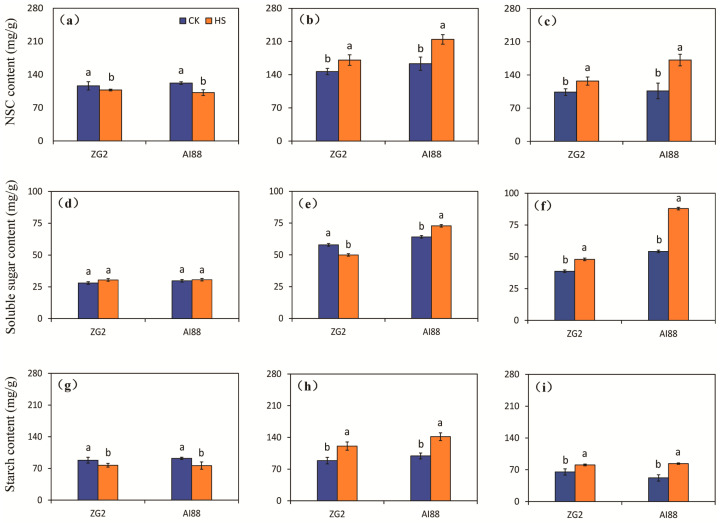
Effect of heat stress on the NSC content, soluble sugar and starch in grains, leaves, and sheath stems under heat stress. The NSC content of the (**a**) grains, (**b**) leaves, and (**c**) sheath stems of ZG2 and AI88, respectively; The soluble sugar content of the (**d**) grains, (**e**) leaves, and (**f**) sheath stems of ZG2 and AI88, respectively; The starch content of the (**g**) grains, (**h**) leaves, and (**i**) sheath stems of ZG2 and AI88, respectively. CK, control; HS, heat stress. A *t*-test was conducted on data to compare the difference between control and heat stress within one cultivar. Different bars and low cased letters within the column chart indicate statistically significant differences between different treatments applied to the same cultivar (*p* < 0.05).

**Figure 4 ijms-25-03171-f004:**
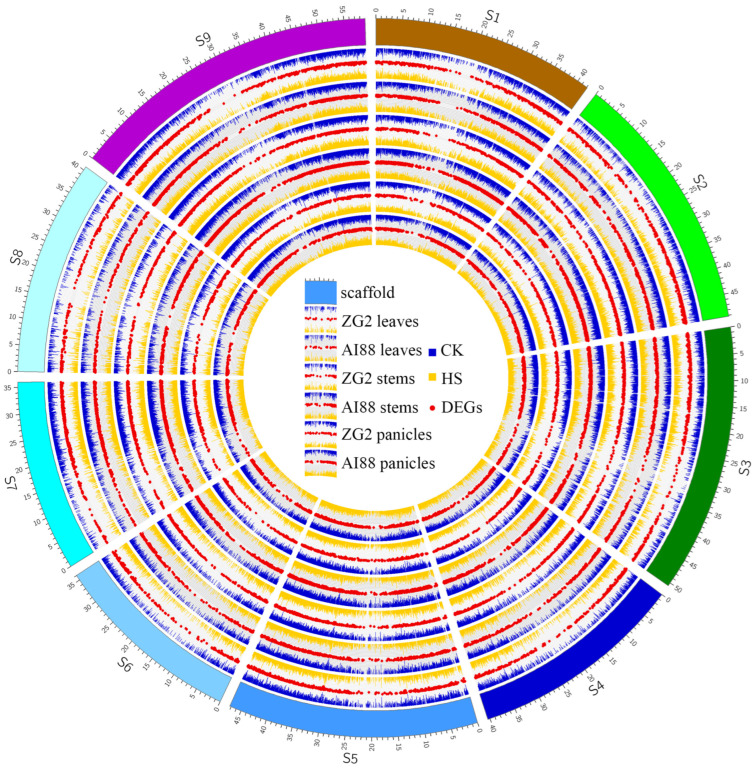
A landscape of the normalized expression level and DEGs of all samples. S1–S9 is the scaffold of the assembled genome, and the ticks on the scaffolds are the physical position. The six expression groups in this figure indicate the three tissues of two varieties, and in each group the blue bars indicate the expression level (log2(FPKM)) of all CK samples; the orange bars indicate the expression level (log2(FPKM)) of HS samples; and red scatters indicate the different expressed genes of each group. CK, control; HS, heat stress; DEGs, the differentially expressed genes.

**Figure 5 ijms-25-03171-f005:**
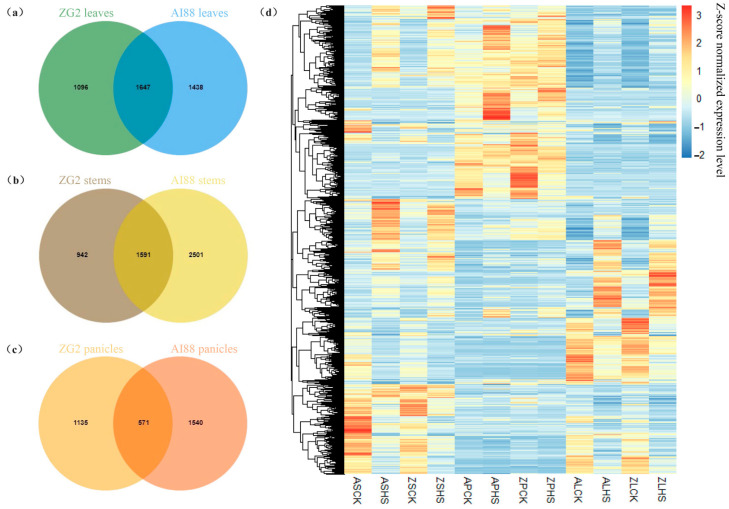
Venn analysis of DEGs in (**a**) leaves, (**b**) stems, and (**c**) panicles. (**d**) Expression level of all DEGs in three tissues of two varieties. The color in the heatmap is the expression level of each gene. “ASCK” is “AI88, sheath and stems, control”, “ASHS” is “AI88, sheath and stems, heat stress ”; “ZSCK” is “ZG2, sheath and stems, control”; “ZSHS” is “ZG2, sheath and stems, heat stress”, “APCK” is “AI88, panicles, control”; “APHS” is “AI88, panicles, heat stress”; “ZPCK” is “ZG2, panicles, control”; “ZPHS” is “ZG2, panicles, heat stress”; “ALCK” is “AI88, leaves, control”; “ALHS” is “AI88, leaves, heat stress”; “ZLCK” is “ZG2, leaves, control”; “ZLHS” is “ZG2, leaves, heat stress”.

**Figure 6 ijms-25-03171-f006:**
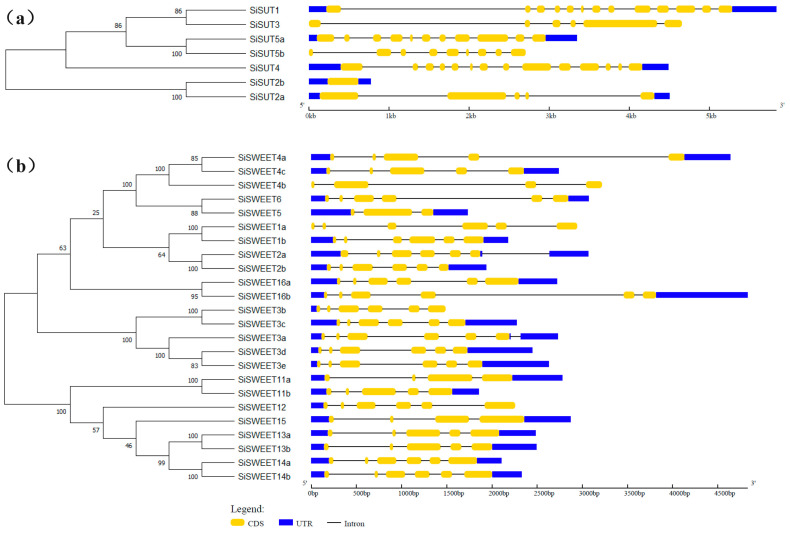
Phylogenetic and gene structure analysis of (**a**) SUT and (**b**) SWEET families. The blue regions indicate the UTR region of the genes, the yellow areas are the coding region of the gene, and the lines indicate the intron region of the genes.

**Figure 7 ijms-25-03171-f007:**
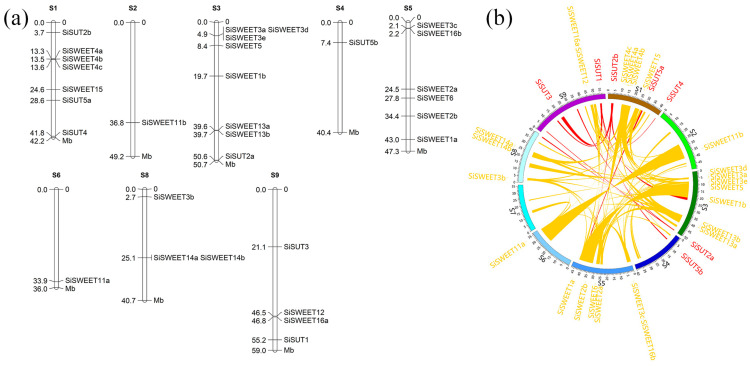
Distribution and duplication analysis of SUT and SWEET family members. (**a**) Distribution analysis of SUT and SWEET families. S1–S6,S8,S9 are the scaffolds of the assembled genome, and the numbers on the right side are the location of genes that were marked on the right side of the scaffolds. (**b**) Segment duplication analysis of SUT and SWEET families. Each link is a segment duplication event in these two families. S1–S9 is the scaffold of the assembled genome; the red ribbons indicate the duplication events of SUTs; and the yellow ribbons indicate the duplication events of SWEETs. Each ribbon is a segment duplication event during the evolution.

**Figure 8 ijms-25-03171-f008:**
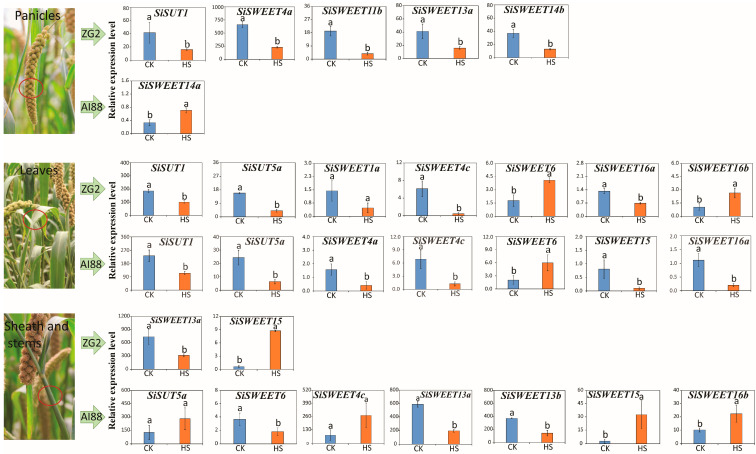
The expression level of different expressed sugar transporters in three tissues of two varieties during heat stress. The three images on the left are the three tissues studied in this experiment, while the left histogram is the expression level of DEGs that belong to each variety. The blue bars indicate the expression level of control, and the orange bars indicate the expression level under heat stress. The red circle in the picture is the place that sampled. Different low cased letters within the column chart indicate statistically significant differences between different treatments applied to the same cultivar (*p* < 0.05).

## Data Availability

The authors confirm that the data supporting the results of this study are available in the article and Appendix A.

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
