# Peer review of "Physiological Investigation and Transcriptome Analysis Reveals the Mechanisms of Setaria italica’s Yield Formation under Heat Stress"

_ijms, 2024, doi:10.3390/ijms25063171_

Round 1

Reviewer 1 Report

Comments and Suggestions for Authors

The present manuscript “Physiological investigation and transcriptome analysis reveals the mechanisms of yield formation under heat stress in Setaria italica” by Hu et al. is the study of the effect of high temperature growth of Setaria italica on some physiological and biochemical parameters to explain reduction in grain yield under these conditions. Overall, the study is well-done and indeed provides a good explanation for the observed effects, namely, in changes of the level of sugar transporters in different tissues that leads to redistribution of carbohydrate content in various plant tissues when growing temperature changes. In addition, the study provides valuable insights into the functioning of sugar transporters in plants.

My main concern is that the authors regard the conditions they used as a “heat stress”. Conditions that are stressful for some plant species may be optimal for others. In order to assess the level of stress in plants used in experiments, there are a large number of indicators. Authors can find them in the literature. These are increased Chl a/Chl b ratio, increased content of violaxanthin cycle pigments, abscisic acid, reactive oxygen species, increased expression level of stress marker genes. The photosynthetic parameters, such as decrease in Fv/Fm, increase of non-photochemical quenching, etc. are also used. Generally speaking, when researchers begin to study the effects of a particular stress, they first expose plants to several different conditions and measure how they affect stress markers. This is done to determine which conditions are stressful indeed, without being lethal. This study shows that in Setaria italica plants at high temperature the chlorophyll content increased with an increase of accumulation of dry matter weight of leaves and stems. The stressful conditions should rather lead to a decrease in chlorophyll content, due to a decrease of the size of light-harvesting photosynthetic antenna. See the recent review Vetoshkina D, Balashov N, Ivanov B, Ashikhmin A, Borisova-Mubarakshina M. Light harvesting regulation: A versatile network of key components operating under various stress conditions in higher plants. Plant Physiol Biochem. 2023 Jan;194:576-588. doi: 10.1016/j.plaphy.2022.12.002. It is also difficult to agree that stressful conditions will lead to the activation of vegetative processes in plants.

Thus, the data indicate that HS for Setaria italica plants are, rather, optimal conditions, than stressful ones. Data on starch increase in vegetative tissues indicate the same. It is known that plants often turn their metabolism to the generation stage in suboptimal or even in stressful conditions. The data of the present research shown that the increase in growing temperature led to a decrease in yield of Setaria italica grain due to a decrease in the number of grains. This is stress for agriculture, but not for plants. Thus, authors should think about why they interpret their data the way they do.

Secondly, the authors should include to the Introduction section the information concerning the sensitivity to high temperatures of Setaria italica plants, if such exist in the literature. If such data are not available, the authors should note that this study is the first of its kind. In the Introduction, the authors currently refer only to publications on wheat, corn, and rice.

The Introduction section should be also added by paragraph with the information about the sugar transporters. It should also include a short story about, what SUT and SWEET families are.

Another common problem with this manuscript is the descriptions of the Figure in their legends, in some cases – with the quality of the Figures.

1.“CK” and “HS” should be designated in each Figure legend. In Figure 4 “HTS” have appeared. What is this?

2. Please, replace all “leaf”, “stem”, etc. and other similar singular forms to plural forms in Figures and throughout the text, because it sounds like laboratory jargon. All studied parameters were not determined in one leaf, plant, etc.

3. Figure 1b. CO2 - should be with a subscript, probably.

4. Figure 2. Percentage in % (OY axis) looks like a nonsense. It seems like, it should be “weight” instead of “percentage”.

5. Figure 3: L. 152-154: “A sharp decrease in the total soluble sugar content was found in the ZG2 leaf, but a clear increase was found in the AI88 leaf (Figure 3e).” It is 17% decrease. I would not say that it is a sharp decrease.

6. Figure 4 arises many questions. I am sorry, I don’t understand it. How can CK, which is the control, HTS (I am not sure, what is this, because “heat stress” is “HS”) and DEG, which is the parameter, be in one row? What are S1-S9 with different colors? It should be specified in Figure legend. Why were ZG2 leaf, AI88 leaf, etc. specified the same way? In the Results the data of Figure 4 are described in just one unclear sentence: “In the end, 2,743 (ZG2, Leaf), 3,085 (AI88, leaf), 2,533 (ZG2, stem), 4,092 (AI88, stem), 1,706 (ZG2, panicle), and 2,111 (AI88, panicle) DEGs were obtained (Figure 4).” Please rewrite this part in scientific language. In the end of what, by the way?

7. The quality of Figure 5 is low. The Figure legend should indicate what the colors of the circles on the diagram and the numbers in them mean, whether the colors in Figure 5a correspond to the colors in Figure 5b. If not, it should be indicated in the Figure legend, what the colors in Figure 5b mean, as well as ASCK, ASHS, etc.

8. Figure 6. “Phylogenetic and gene structure analysis of SUT (a) and SWEET (b) families” sounds like a nonsense. At that, the name of Figure S2 is “Phylogenetic analysis of the SUT and SWEET family members”, which makes sense.

9. Figure 7 legend also obviously needs a more detailed and clear description (S1-S9, colors), the yellow inscriptions in Figure 7b are not readable.

10. Figure 10. The inscriptions are not readable

All abbreviations (NSC, DEG, SUT, SWEET, etc.) should be specified at the first mention and not in the Materials and Methods section at the end of the manuscript.

L. 108-110: The sentence “The effect of heat stress on the seed-setting rate of the ZG2 and AI88 plants was less than the effect on the controls (Figure S1b).” is unclear. I see that “there were no effect of high temperature on the seed-setting rate for both ZG2 and AI88 plants”.

In Materials and Methods, the seed-setting rate parameter calculation should be added.

Comments on the Quality of English Language

There are expressions in the text that sound like a laboratory jargon style of presenting results. It is used in oral speech to simplify the expressions, but it is unacceptable in scientific text.

-         All singular forms, such as “leaf”, “stem”, etc. and other similar expressions should be replaced to plural forms in Figures and throughout the text. All studied parameters were not determined in one leaf, plant, etc.

-         In the end, 2,743 (ZG2, Leaf), 3,085 (AI88, leaf), 2,533 (ZG2, stem), 4,092 (AI88, stem), 1,706 (ZG2, panicle), and 2,111 (AI88, panicle) DEGs were obtained (Figure 4)…

-         The effect of heat stress on the seed-setting rate of the ZG2 and AI88 plants was less than the effect on the controls (Figure S1b).

-         Figure 6. “Phylogenetic and gene structure analysis of SUT (a) and SWEET (b) families”…

Reviewer 2 Report

Comments and Suggestions for Authors

The paper entitled “Physiological investigation and transcriptome analysis reveals the mechanisms of yield formation under heat stress in Setaria italicashows a potential mechanism of temperature on crop formation of foxtail millet. 

After reading the text, I had a few comments and suggestions. 

Maior comments:

1.     The authors focused on temperature resistance and related parameters. However, global warming is, of course, associated with increased temperatures, but it also results in drought. The authors should discuss water supply as one of the factors. Water affects growth and transpiration, but also photosynthesis, biosynthesis of metabolites, and their transport. The water requirements of the tested species should be known. Field cultivation conditions must be given and how they could affect the tested varieties.

2.     Balance in the number of citations should be maintained. There are 35 references in the Introduction, 9 in the Methodology, and only 12 in the Discussion. The discussion is a key part of the scientific work.

3.     I couldn't find anywhere what the sample number was in the main experiment. Statistics and captions for Figures 1-3 state n=3. Lines 341-342 as well. If this is true, it is definitely not enough for a biological experiment where the influence of the environment and variability is obvious. What does mean “biological and technical replicates”?

Minor comments: 

1.     Are the differences between varieties temperature-resistant or something else? Drought resistance, growth rate, etc.

2.     The Latin names of the plants should also be written in italics. Abstract, Keywords, line 90, in some places in the methodology.etc.

3.     Line 44. S. italica not Italica.

4.     Fig. 1,2,3 "Different letters within the same column..." These are not columns, but bars.

5.     Fig. 7 must be larger.

6.     Figure 8 needs to be larger with better resolution. No, it is possible to read anything even in PDF. Figure capitation must be completed as well.

7.     Formatting of references is needed. Latin names should be in italics and with a capital letter of the genus. Ref. 4. The initial of the second author is missing.

8.     There are some typos in the text, so please read it carefully. 

Reviewer 3 Report

Comments and Suggestions for Authors

Round 2

Reviewer 1 Report

Comments and Suggestions for Authors

Dear authors! I attached file with responses (brown) to your responses. 

Reviewer 2 Report

Comments and Suggestions for Authors

“In this study, three biological replicates and three technical replicates were performed in all experiments”

I don't know if the authors understood my question. The answer provided does not dispel my doubts.

The main problem according to what remains for me is the number of repetitions. 3 biological replicates (2 varieties, tested and control for each), i.e., 3 plants divided into flowers (male and female), root, fruit, leaves. 3 technical repetitions, i.e. three measurement repetitions for each determination.

Translate this into animal studies: 3 study mice and 3 control mice. Different organs taken from them and 3 repetitions of each biochemical determination. No one would accept such results.

Please explain this or repeat the experiment.

Author Response

Thank you for your professional comments. We didn’t explain it clearly and sorry for that. We have revised this section in the manuscript of 4.1 and 4.2, and added more information to make it clearer and accurate.

In this study, we have four groups, including control of ZG2, heat stress of ZG2, control of AI88, and heat stress of AI88. Each group have 15 pots, and each pot contained 5 plants. When study processed, three plants were selected randomly as one measure repeat, and each group we processed at least three measurement repetitions. At the same time three measurement repetitions for each determination were carried in the experiment.

For the measurement of chlorophyll concentration, MDA content, and seed-setting rate, three randomly selected plants were used as one repeat, each group we processed five measurement repetitions. And three measurement repetitions for each determination were carried.

As for the measurement of net photosynthetic rate, six randomly selected plants were used for the data acquisition in every group (one plant was set as one measure repeat), and three technical repetitions were also used for each determination.

Thanks again for your patience and professional work.

Round 3

Reviewer 1 Report

Comments and Suggestions for Authors

Dear Authors!

You corrected everything, according to my comments, except "“CK” and “HS” in the Figure legends as: CK, control; HS, heat stress". It was done only for Figure 4 legend. It should be in every Figure legend. It seems like it is very easy to do. 

Author Response

Thank you for your professional comments. We have revised the manuscript according to your suggestions.

Reviewer 2 Report

Comments and Suggestions for Authors

Thank you for your response. Now i am sure what was performed.

Author Response

Thank you for your professional works. And we checked all the cited references, language, and other details carefully in the manuscript.